# Pharmacogenetic Expression of CYP2C19 in a Pediatric Population

**DOI:** 10.3390/jpm12091383

**Published:** 2022-08-26

**Authors:** Marie Josette Déborah Pierre-François, Vincent Gagné, Ivan Brukner, Maja Krajinovic

**Affiliations:** 1Department of Pharmacology and Physiology, University of Montréal, Montreal, QC H3C 3J7, Canada; 2Department of Pediatrics, University of Montreal, Montreal, QC H3T 1C5, Canada; 3Lady Davies Research Institute, Montreal, QC H3T 1E2, Canada; 4CHU Sainte-Justine Research Center, Montreal, QC H3T 1C5, Canada

**Keywords:** CYP2C19, metabolizer, voriconazole, clopidogrel, anti-depressants, proton pump inhibitors, pediatrics, trough plasma concentration, genotyping

## Abstract

Genetic variability in CYP2C19 may be associated with both lack of efficacy and toxicity of drugs due to its different metabolic status based on the presence of particular alleles. This literature review summarizes current knowledge relative to the association or treatment adaptation based on *CYP2C19* genetics in a pediatric population receiving drugs metabolized by CYP2C19, such as voriconazole, antidepressants, clopidogrel and proton pump inhibitors. Additionally, we also presented one of the approaches that we developed for detection of variant alleles in the CYP2C19 gene. A total of 25 articles on PubMed were retained for the study. All studies included pediatric patients (age up to 21 years) having benefited from an assessment of CYP2C19. CYP2C19 poor and intermediate metabolizers exhibit a higher trough plasma concentration of voriconazole, and PPIs compared to the rapid and ultra-rapid metabolizers. The pharmacogenetic data relative to CYP2C19 and clopidogrel in the pediatric population are not yet available. CYP2C19 poor metabolizers have a higher trough plasma concentration of antidepressants compared to the rapid and the ultra-rapid metabolizers. Modification of allele-specific PCR through the introduction of artificial mismatch is presented. CYP2C19 genotyping remains a powerful tool needed to optimize the treatment of children receiving voriconazole, PPIs, and anti-depressants.

## 1. Introduction

The CYP2C19 gene is composed of nine exons coding for a 490 amino acid protein and is one of four genes of the CYP2C subfamily located on chromosome 10q23.33 (CYP2C8, CYP2C9, CYP2C18, and CYP2C19) [1]. The *CYP2C19* gene is highly polymorphic. Alleles are categorized into functional groups including those with normal function (*CYP2C19*1*), decreased function (*CYP2C19*9*), no function (*CYP2C19*2* and **3*), and increased function (*CYP2C19*17*) [1]. The combination of inherited alleles determines a person’s diplotype. Metabolizer status such as normal or extensive (EM), intermediate (IM), rapid (RM), and ultra-rapid metabolizers (UM) is defined by different diplotypes [1] (Table 1).

It is now recommended to use pharmacogenomics (PGx) to help therapeutic decisions through two different models: the point-of-care model, also known as the reactive model; and the pre-emptive model. The reactive model involves only one or more targeted gene–drug combinations and is usually guided by a clinical assessment at the time of prescribing or in response to an emerging or past adverse event, including lack of therapeutic benefit. Alternatively, the preemptive PGx model is an active approach that addresses potential drug therapies using genotyping strategies involving the testing of multiple pharmacogenes independent of an individual’s drug history [2].

Genetic variation in CYP2C19 impacts the metabolism of many drugs and has been associated with efficacy and safety issues for several commonly prescribed drugs [1]. The CYP2C19 enzyme contributes to the metabolism of many clinically used drugs, including clopidogrel, voriconazole (VCZ), proton pump inhibitors (PPIs), anti-depressants, carisoprodol, and diazepam [1]. In 2009, the Clinical Pharmacogenetics Implementation Consortium (CPIC), a shared project between Pharmacogenomics Knowledge Base (PharmGKB) and the National Institutes of Health (NIH), was formed to provide pharmacogenetic clinical practice guidelines. To date, the CIPC has published 23 guidelines covering 19 genes, including CYP2C19 and 46 drugs in several therapeutic areas, based on previous research studies [3]. However, while the literature on the CYP2C19–drug associations or implementation is largely available for the adult population, less is known for children.

This literature review summarizes current knowledge relative to the association or treatment adaptation based on *CYP2C19* genetics in a pediatric population receiving drugs such as voriconazole, clopidogrel, PPIs, and antidepressants.

**Table 1 jpm-12-01383-t001:** The predicted CYP2C19 phenotype based on genotype.

Predicted Phenotype	Genotype	Examples of *CYP2C19* Diplotypes
CYP2C19 ultra-rapid metabolizer	An individual carrying two increased function alleles	**17/*17*
CYP2C19 rapid metabolizer	An individual carrying one normal function allele and one increased function allele	**1/*17*
CYP2C19 normal metabolizer	An individual carrying two normal function alleles	**1/*1*
CYP2C19 likely intermediate metabolizer	An individual carrying one normal function allele and one decreased function allele or one increased function allele and one decreased function allele or two decreased function alleles	**1/*9*, **9/*17*, **9/*9*
CYP2C19 intermediate metabolizer	An individual carrying one normal function allele and one no function allele or one increased function allele and one no function allele	**1/*2*, **1/*3*, **2/*17*, **3/*17*
CYP2C19 likely poor metabolizer	An individual carrying one decreased function allele and one no function allele	**2/*9*, **3/*9*
CYP2C19 poor metabolizer	An individual carrying two no function alleles	**2/*2*, **3/*3*, **2/*3*
Indeterminate metabolizer	An individual carrying one or two uncertain function alleles	**1/*12*, **2/*12*, **12/*14*

Lee et al. (2022) [4]. A complete list is available on https://www.pharmgkb.org/page/CYP2C19RefMaterials (accessed on 28 April 2022).

## 2. Selection of the Studies

A literature review was performed through PubMed to address how CYP2C19 metabolizer status influences the occurrence of adverse effects or treatment failure in children aged 0–21 years receiving voriconazole, clopidogrel, PPIs, and anti-depressants. It was limited to systematic reviews, meta-analyses, reviews, evaluation reports, and guidelines.

The inclusion criteria included studies published between 2010 and 2022 in English or French on the pediatric population (age up to 21 years) having addressed CYP2C19 genotyping. The aim of these studies was to predict treatment failure and the occurrence of adverse effects in relation to CYP2C19 genotypes. Exclusion criteria were articles on adult patients and articles dealing with cytochromes other than CYP2C19. A total of 79 articles were found in PubMed but, in the end, 39 articles were retained. The Boolean operators used were: CYP2C19 and pediatrics, CYP2C19 and voriconazole, CYP2C19 and clopidogrel, CYP2C19 and proton-pump inhibitors, CYP2C19 and antidepressants, CYP2C19 and children, and CYP2C19 and genotyping. Data were sought mainly for the following variables: age, primary diagnosis, metabolizer status, use of medications (voriconazole, clopidogrel, PPIs, anti-depressants), occurrence of adverse effects, treatment failure, elevated or decreased minimum active drug concentration.

## 3. Study Descriptions

### 3.1. Voriconazole and CYP2C19 Metabolizer Status

Voriconazole (VCZ) is a triazole antifungal agent that inhibits fungal cytochrome P450-mediated demethylation of 14 alpha-lanosterol, an essential phase in fungal ergosterol biosynthesis. Voriconazole has a broad spectrum of activity and acts non-exhaustively on the following species: *Aspergillus*, *Candidiasis*, *Fusarium*, and *Sedosporium*. Voriconazole is currently recommended as a first-line treatment for acute invasive aspergillosis, as a treatment for infections caused by *Fusarium* and *Scedosporium*, and as a prophylactic agent in children undergoing hematopoietic cell transplantation (HCT) [5]. The hepatic enzymes CYP2C19, CYP2C9, and CYP3A4 are responsible for the metabolism of voriconazole. CYP2C19 is primarily responsible for the conversion of voriconazole to its major inactive metabolite, voriconazole-*N*-oxide, accounting for approximately 72% of plasma metabolites. CYP2C19 polymorphisms account for most of the interindividual variability of voriconazole [6]. Despite the widespread use of voriconazole, optimization of its treatment in an individual becomes difficult because of its large interindividual variability [7]. Voriconazole has a narrow therapeutic index. Numerous studies have shown that high steady-state plasma concentrations (>5 mg/L) are associated with clinical adverse effects, whereas inadequate concentrations (<1.5 mg/L) are likely to result in treatment failure [6]. CYP2C19 metabolizer status significantly influences voriconazole plasma concentrations.

Zhao et al. (2021) [6] performed a non-interventional retrospective clinical study aimed at investigating the optimal maintenance dose as well as factors affecting trough VCZ concentration. This study was conducted in 94 children in whom 145 voriconazole trough concentrations were available. The primary diagnosis was hematological malignancy followed by respiratory infection and bacteremia. Final multivariate analysis revealed that weight, drug dose before sampling, direct bilirubin, urea nitrogen, and CYP2C19 genotypes were factors influencing the trough concentration of voriconazole explaining 36.2% of the variability in drug concentration [6]. Narita et al. (2013) [8] conducted a retrospective study of 37 Japanese children genotyped for CYP2C19*2, *3, and *17 and analyzed their relation to previously measured plasma voriconazole concentrations. The authors concluded that poor or intermediate metabolizers had trough concentrations above 5 μg/mL, which was significantly higher compared to normal and ultra-rapid metabolizers. Two patients with high plasma voriconazole concentrations experienced severe side effects: inappropriate anti-diuretic hormone secretion syndrome and cardiac toxicity [8]. For Tian et al. (2021) [9], the minimum concentration of VCZ was also higher in CYP2C19 IMs and PMs compared with CYP2C19 EMs [9].

Dose adjustment based on CYP2C19 genotype may be useful during voriconazole therapy [8]. As a matter of fact, the final model simulation by Takahashi et al. (2021) [5] suggested the following doses according to metabolizer status and weight to achieve target trough concentrations of 1.5–5.0 mg/L: for normal metabolizers: 16 mg/kg (15 kg weight), 12 mg/kg (15–30 kg weight), or 10 mg/kg (30 kg weight); doses were 33–50% lower for poor and intermediate metabolizers and 25–50% higher for rapid and ultra-rapid metabolizers [5]. Furthermore, Hicks et al. (2020) [10] conducted a prospective study to determine the impact of a higher prophylactic dose of voriconazole (300 mg) given twice daily to CYP2C19 rapid metabolizers in decreasing the incidence of subtherapeutic trough concentrations without exacerbating CYP2C19-induced toxicities. CYP2C19 rapid metabolizers were recommended to receive interventional voriconazole 300 mg twice daily; ultra-rapid metabolizers were recommended to avoid voriconazole; and others were recommended to receive the standard prophylactic dose of 200 mg twice daily. Subtherapeutic concentrations were prevented in 83.8% of CYP2C19 rapid metabolizers receiving interventional dosing versus 46.2% receiving standard dosing [10]. One year later, Garcia-Garcia et al. (2021) [11] reported analyses of 28 immunocompromised pediatric patients in whom preemptive CYP2C19 genotyping and therapeutic VCZ monitoring was performed. The final objective was to compare its results with the results obtained by Hicks et al. and those expected in their PGx-based VCZ assay simulation. Plasma trough concentrations were measured by immunoassay until target concentrations (1–5.5 μg/mL) were reached. Standard dose modifications were indicated in 29% of patients. Patients with CYP2C19*1/*1, *1/*2, and *2/*17 (CYP2C19 NMs and IMs) received standard starting doses, whereas CYP2C19*1/*17 and *17/*17 patients (CYP2C19 RMs and UMs) had increased starting doses. No PMs were found in the cohort. Sixteen patients (57.14%) reached the minimum target concentrations of VCZ in the first measurement after the initial PGx-based dose. Combined genotyping and the drug therapeutic monitoring strategy achieved target concentrations during treatment and prophylaxis in 90% of CYP2C19 NMs/IMs; and in 100% of RMs/UMs [11]. Wang et al. (2021) [12] used a PK model to optimize the voriconazole dosing regimen in children with severe disease. A total of 99 children below 14 years were included in the study. The Bayesian estimate suggests that the dose-normalized concentration and the total exposure were significantly different between EM and PM patients. Maintenance doses were reduced by approximately 30–40% in PM patients compared to EM patients. For children who are ultra-rapid metabolizers aged 2 years, the maintenance doses were higher [12]. Ultimately, CYP2C19 genotyping can guide prophylactic dosing of voriconazole and thus reduce the risk of subtherapeutic trough concentrations promoting fungal infections. The inclusion of CYP2C19 genotyping and therapeutic VCZ monitoring in clinical practice may help achieve therapeutic target concentrations.

The drug package leaflet for voriconazole recommends a weight-based dose of 9 mg/kg per 12 h by intravenous (IV) or oral (PO) route for children aged 2–12 years. The loading dose is 6 mg/kg per 12 h IV for patients older than 12 years [13]. It has also been suggested that 5–7 mg/kg is adequate in children under 2 years of age [9]. To date, the optimal dose in pediatric patients has not been fully established. Tian et al. (2021) [9] found that for CYP2C19 UM or EM metabolizers, patients younger than 12 years and older than 12 years required doses of 6.53 and 3.95 mg/kg twice daily, respectively, to achieve a therapeutic trough concentration. On the other hand, for CYP2C19 PMs or IMs, patients younger than 12 years and older than 12 years required doses of 5.75 and 4.23 mg/kg twice daily, respectively, less than the doses required for UM or EM metabolizers [9]. For Chen et al. (2022) [13], the median daily dose of voriconazole required to reach the therapeutic range was: 20.8 mg/kg (range, 16.2–26.8 mg/kg) for the NM group, 18.2 mg/kg (range, 13.3–21.8 mg/kg) for the IM group, and 15.2 mg/kg (range, 10.7–19.1 mg/kg) for the PM group [13].

Some confounders were not considered in all studies, such as the use of concomitant medications which is particularly applicable for the use of voriconazole in critically ill children. As demonstrated in the study by Tian et al. (2021) [9], omeprazole increases the concentration of VCZ by competitive inhibition of CYP2C19 and rifampicin decreases the concentration of VCZ by enzyme induction [9]. Rifamycin is a potent enzyme inducer that increases CYP2C19 activity and therefore enhances the elimination of certain drugs such as VCZ resulting in decreased efficacy. In clinical experience, when patients were concomitantly treated with rifamycin, VCZ trough levels remained low for up to 14 days after VCZ discontinuation. Accordingly, in this study there was a significant difference in VCZ trough levels between the rifamycin and no rifamycin co-administration groups. Furthermore, omeprazole can increase VCZ trough concentration because it is a competitive inhibitor of CYP2C19. Indeed, among five PPIs, omeprazole was shown to have a significant effect on VCZ concentration compared to patients without PPIs. Another confounding factor would be inflammation, which is not accounted for in all studies with voriconazole. As pointed out by Takahashi et al. (2021) [5], patients may be prone to inflammation after chemotherapy. Inflammation can suppress the activities of several enzymes including CYP2C19 [5]. The results are consistent across the studies carried out on voriconazole. Poor and intermediate CYP2C19 metabolizers receiving voriconazole have a higher trough concentration than rapid and ultra-rapid metabolizers. Indeed, the metabolic status of CYP2C19 is critical in adjusting the dosage of voriconazole. CYP2C19 genotyping is therefore recommended for therapeutic or prophylactic use of VCZ in children.

Refer to Table 2 summarizing the studies cited above.

### 3.2. Clopidogrel and CYP2C19 Metabolizer Status

Clopidogrel is a thienopyridine prodrug that requires hepatic biotransformation to form an active metabolite that selectively and irreversibly inhibits the P2Y12 receptor and thus platelet aggregation. Only 15% of the prodrug is available for conversion to active ingredient; the remaining 85% is hydrolyzed by carboxylesterase-1 (CES1) into inactive forms [4]. Clopidogrel is commonly prescribed to reduce the risk of myocardial infarction and stroke in patients with acute coronary syndromes following percutaneous coronary intervention. Despite the availability of newer, more potent agents (prasugrel and ticagrelor), clopidogrel remains the most commonly prescribed antiplatelet drug in North America for the above indications [4]. Clopidogrel is not commonly used in the pediatric patient population. It is used in the case of Kawasaki disease [14] and it can also be used before a heart transplant [15].

The conversion of clopidogrel to its active metabolite requires two sequential oxidative steps involving several CYP450 enzymes (e.g., CYP1A2, CYP2B6, CYP2C9, CYP2C19, and CYP3A4/5). Nevertheless, in both steps, CYP2C19 has the greatest contribution of all these enzymes [4]. Genetic variation in CYP2C19 does not explain all the variability in response to clopidogrel. Some studies have implicated variants in other genes associated with clopidogrel response, such as ABCB1, B4GALT2, CES1, CYP2B6, CYP2C9, P2RY12, and PON1. However, these studies have not been consistently replicated, and CYP2C19 is the most validated genetic determinant of response to clopidogrel [4].

The active metabolite of clopidogrel is reduced in CYP2C19 poor and intermediate metabolizers while the concentration of the active metabolite is higher in rapid and ultra-rapid metabolizers. The clinical data on which the recommendations are based have been obtained from studies in adults [4]. To date, there are no studies on the pharmacogenetics of clopidogrel in the pediatric population. In view of the well-characterized pharmacokinetics of this gene–drug interaction and the presence of fully mature CYP2C19 enzyme activity after 2–3 months of age, it is reasonable to extrapolate adult recommendations to pediatric patients if necessary. However, studies should be undertaken to investigate the pharmacogenomics of clopidogrel in the pediatric population [4].

### 3.3. PPIs and CYP2C19 Metabolizer Status

Six PPIs are currently approved in the United States, including omeprazole, lansoprazole, dexlansoprazole, pantoprazole, rabeprazole, and esomeprazole. PPIs exert their pharmacological action by irreversibly inhibiting the H+/K+ ATPase proton pump in gastric parietal cells, and thereby inhibiting gastric acid secretion [16].

CYP2C19 is responsible for more than 80% of the metabolism of esomeprazole, lansoprazole, and pantoprazole. Previous pediatric studies have suggested little effect of CYP2C19 variation on PPIs [17,18]. However, recent data have supported a role of the CYP2C19 genotype in PPI response, similar to that seen in adults [16]. In rapid metabolizers of CYP2C19, esomeprazole, lansoprazole, and pantoprazole are inactivated more rapidly. In contrast, poor metabolizers have approximately twice the exposure to these drugs [16].

Franciosi et al. (2018) [19] conducted a retrospective cohort study to confirm the relationship between gastroesophageal reflux disease (GERD) refractory to PPIs and genetic CYP2C19 variants. These children were split into CYP2C19*17 allelic carriers and non-carriers and correlated to pH probe acid exposure measured by gastroscopy. Compared to controls, children carrying the CYP2C19*17 alleles demonstrated a longer duration of exposure to an acidic pH [19]. Since CYP2C19 inactivates PPIs, genetic variants that increase CYP2C19 function may decrease PPI exposure and infections. The infectious pathologies linked to the use of PPIs would be secondary to a reduction in gastric acidity and the resulting dysbiosis of the gastric microflora, thus, causing colonization by pathogenic microbes [20]. A retrospective study including children aged 0–36 months at the time of exposure to PPIs was conducted by Bernal et al. (2019) [20]. The objective was to test the hypothesis that CYP2C19 metabolizing groups are associated with rates of infectious events in children exposed to PPIs. The group of normal CYP2C19 metabolizers (*n* = 267; 40%) had a higher infection rate than the MR/MU (*n* = 220; 33%). No difference was observed between poor or intermediate metabolizers (*n* = 183; 27%) and normal metabolizers. The latter observation might be related to the population size of poor/intermediate metabolizers, which is much smaller than that of normal metabolizers. In the multivariate analysis of NM and MR/UM adjusting for age, sex, PPI dose, and comorbidities, CYP2C19 metabolizer status was a significant risk factor for infectious events with higher infection rates in normal metabolizers compared to those with rapid/ultra-rapid metabolizing capacity [20].

Moreover, Mougey et al. (2019) [21] conducted a prospective study whose objective was to determine the influence of genetic variation of two genes including CYP2C19 on the treatment of eosinophilic esophagitis by PPIs. Of 92 patients examined, 57 (62%) were patients with PPI-responsive eosinophilic esophagitis and 35 (38%) were patients with non-PPI-responsive disease. In children who received a dose of PPI between 1.54 and 2.05 mg/kg/day (the estimated dose to reach 80 mg per day), a binary logistic regression model showed that the presence of the CYP2C19 *17 allele was associated with a higher risk of PPI-insensitive eosinophilic esophagitis [21]. In conclusion, normal, poor, and intermediate CYP2C19 metabolizers are associated with a higher risk of infections. In addition, rapid CYP2C19 metabolizers are associated with treatment failure in eosinophilic esophagitis. All presented studies are in favor of genotyping tests in patients receiving PPIs.

Refer to Table 3 summarizing the studies cited above.

### 3.4. Anti-Depressants and CYP2C19 Metabolizer Status

Anxiety and depressive disorders are the most common mental health problems in children and adolescents and constitute one of the major health care burdens for people under the age of 18 [22]. As their prevalence increases, clinicians are challenged to find effective early treatments to prevent disease progression. Five antidepressants that belong to selective serotonin reuptake inhibitor (SSRI) or serotonin-norepinephrine reuptake inhibitor (SNRI) classes have FDA indications for patients under 18: escitalopram for major depression; fluoxetine for major depression and OCD (obsessive-compulsive disorder); fluvoxamine, sertraline for OCD; and duloxetine for generalized anxiety disorder [22]. SSRIs are primarily metabolized by hepatic cytochrome P450 enzymes, including CYP2D6 and CYP2C19 [22]. Sertraline is metabolized by CYP2D6 and CYP2C19, although its pharmacokinetics appear to be primarily influenced by CYP2C19 variants. Few studies have focused solely on CYP2D6 or CYP2C19 genotype and its association with pharmacokinetic parameters or treatment outcomes of tricyclic antidepressants (TCAs) in pediatric patients [23]. Tertiary amines, such as amitriptyline, are primarily metabolized by CYP2C19 to demethylated metabolites, also called secondary amines, such as nortriptyline [23]

Although an arsenal of psychotropic drugs is available, there is heterogeneity in treatment response and drug tolerance attributed to different factors including genetics. Genetics is responsible for approximately 40% of the variability in response to antidepressants in major depression [22].

Ariefdjohan et al. (2021) [24] conducted a retrospective study of pediatric patients aged 1–22 years receiving care for a psychiatric disorder at a large urban hospital between January 2015 and November 2016. The objective of this study was to describe trends and clinical experiences in the application of commercial pharmacogenetic testing in pediatric patients with neuropsychiatric disorders. A total of 450 patients (12.1 ± 4.3 years) diagnosed with anxiety disorder, attention deficit hyperactivity disorder, developmental disorders including autism and/or mood disturbances were tested and 435 of them received medication. By comparing the data before and after carrying out the pharmacogenetic tests, the total number of psychotropic prescriptions was reduced by 27.2% and the number of drugs prescribed with serious gene–drug interactions decreased from 165 to 95 (11.4% to 8.9% of total drugs prescribed). About 40% of usable genetic annotations were related to CYP2CD6 and CYP2C19. Among SSRIs, paroxetine (54%) had the highest proportion of patients with side effects due to CYP2C19 metabolizer status, followed by fluvoxamine (42%), then fluoxetine (29%) and among SNRIs duloxetine (42%) topped the list followed by venlafaxine (20%). In contrast, desvenlafaxine (an SNRI) had no side effects attributable to CYP2C19 metabolizing status because it was metabolized to lesser extent by CYP2C19. For bupropion (a DNRI: Dopamine and Noradrenaline Reuptake Inhibitor), 19% of patients in the study cohort had side effects attributable to CYP2C19 [24].

Using pharmacokinetic (PK) models in adolescents, Strawn et al. (2019) [25] performed SSRI dosing CYP2C19-dependent modeling. The objective was to assess the impact of CYP2C19 metabolizers on SSRI exposure and peak concentration; and finally, to determine dosing strategies based on pharmacogenomics. Compared to normal CYP2C19 metabolizers treated with escitalopram or sertraline, Cmax and area under the curve (AUC) were higher in poor and intermediate metabolizers than in patients with increased CYP2C19 activity, although the magnitude of these differences was more pronounced for escitalopram than for sertraline. For escitalopram, PMs need 10 mg/day and UMs needed 30 mg/day to reach an exposure equivalent to 20 mg/day usually seen in a normal metabolizer. Twice daily escitalopram dosing was required in UMs to achieve comparable levels of depression and exposure to NMs [19]. For sertraline, to achieve AUC and Cmax like NMs receiving 150 mg/day, PM needed 100 mg/day, whereas 200 mg/day was needed in rapid and ultra-rapid metabolizers [25].

Aldrich et al. (2019) [26] conducted a retrospective study of electronic medical record data from 263 young people under the age of 19 with anxiety and/or depressive disorders receiving escitalopram or citalopram and having undergone routine clinical examinations for genotyping of the CYP2C19. Poor CYP2C19 metabolizers had more adverse effects than ultra-rapid metabolizers, including serotonin syndrome and faster weight gain. Treatment was discontinued in most poor metabolizers treated with escitalopram compared to normal metabolizers. In addition, ultra-rapid metabolizers responded more quickly to escitalopram and their subsequent hospital stays were shorter [26].

There is a heterogeneity of the results regarding the antidepressants. The majority of studies found that poor metabolizers have a higher plasma concentration of antidepressants and are, therefore, more prone to adverse effects compared to remaining metabolizing groups. Contrary to this observation, Rossow et al. (2020) [27] have found that normal CYP2C19 metabolizers have significantly more adverse effects compared to poor metabolizers and intermediate metabolizers. Sertraline AEs were more frequent in NMs than in PMs or IMs with a hazard ratio (HR) of almost 2 in the unadjusted and adjusted analysis. For (es)citalopram, more AEs were observed also in NMs than in PMs and IMs but without statistically significant differences. Rossow’s results are surprising because they do not reflect expected results based on metabolic status [27]. Rossow et al. suggest a specific interaction between the metabolizing status of children or adolescents and sertraline or (es)citalopram. Their study population consisted entirely of adolescent girls. It is possible that there is sex-dependent altered expression of cytochrome P450 enzymes during adolescence that affects the metabolic pathway of sertraline and escitalopram. To that end, there is evidence that estrogen inhibits the expression of CYP2C19 [27].

Refer to Table 4 summarizing the studies cited above.

## 4. Genotyping Assays for CYP2C19

As reviewed above, there is an increasing number of recommendations on how to adjust the treatment based on the CYP2C19 genotypes to reduce the risk of treatment failure or adverse drug events. Simple, accurate, time- and cost-effective tests are essential for diagnosis, screening, and research purposes. This need is even more potentiated with initiatives to develop pharmacogenetic implementation networks, which can more broadly involve physicians, health professionals, and community pharmacists. Many genotyping approaches for CYP2C19 have been developed by academic laboratories or pharmaceutic companies [28,29,30]. Different methods are used as summarized in [2], such as polymerase chain reaction–restriction fragment length polymorphism (PCR-RFLP), TaqMan assay, allele-specific PCR (ASP-PCR), high-resolution melting (HRM), pyrosequencing, Mass ARRAY, DNA direct sequencing, and amplification-refractory mutation system–polymerase chain reaction (ARMS-PCR). All these methods have their respective advantages and disadvantages. ASP-PCR does not require expensive equipment, and is less expensive than other methods [31]. It is accessible to minimally equipped laboratories and is appropriate for testing one patient at a time. It allows discrimination of genotypes using standard PCR conditions in which a common reverse primer and two forward allele-specific primers, delineating two alleles of the given SNP, are used in two parallel PCR reactions [32]. PCR products can be simply analyzed using qPCR technology or agarose gel electrophoresis. Despite these advantages, the genotype differentiation by AS-PCR is not always easily achieved. PCR specificity in terms of discrimination between major and minor alleles or wild-type and mutant alleles can be largely improved by the introduction of a destabilizing mismatch in the allele-specific primers [33]. It is based on the idea that two mismatches, diagnostic and artificial, are less likely to be amplified than only one artificial mismatch, present in the case of correct genotype. Our preliminary analyses indicate that the introduction of the artificial mismatch (simple replacement by a complementary nucleotide) at the fourth position from the mismatch discriminating two alleles (allele-specific nucleotide at a diagnostic position), has an excellent specificity in discriminating the genotypes in most of the cases. The example of such primer design and resulting PCR amplification to identify three SNPs underlying CYP2C19 *2, *3, and *17 alleles is presented in Figure 1 and Table 5. Our method gave accurate genotyping results with 100% concordance when compared to known CYP2C19 genotypes of 21 Coriell lymphoblastoid cell lines samples derived from 3 Asians and 18 individuals of northern and western European ancestry.

## 5. Conclusions

CYP2C19 genotypes are associated with the pharmacokinetic changes in voriconazole, PPIs, and anti-depressants in pediatric populations. Studies of clopidogrel are needed as the indication remains for Kawasaki disease and prevention of ventricular assist device thrombosis in children. Recommendations based on some of these clinical experiences already exist; an update is needed to reflect more recent clinical experiences in pediatric populations. Clinicians could thus consider the recommendations to optimize treatment in children. Larger prospective studies are needed to amplify the pediatric data on CYP2C19 pharmacogenetics. Simple genotyping assays applicable to the analyses of single patients at the time are suited for such pharmacogenetic studies.

## Figures and Tables

**Figure 1 jpm-12-01383-f001:**
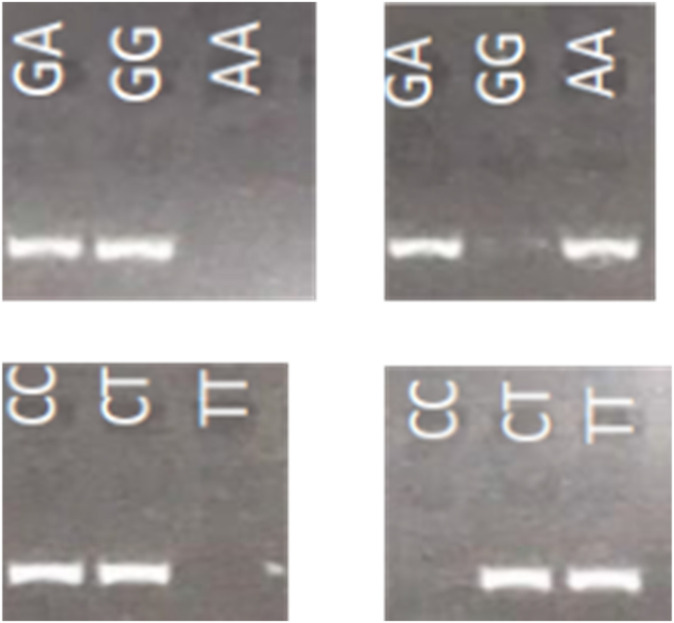
Allele-specific PCR for CYP2C19 alleles. Illustrative example of genotypes obtained by modification of allele-specific PCR through destabilizing mismatches, distinguishing 3 possible genotypes of **CYP2C19** polymorphisms whose minor alleles define variant *2 (**upper panels**) and *17 (**lower panel**) with decreased and increased enzyme activity, respectively. (**Left panels**), PCR specific for major alleles; (**Right panel**), PCR specific for minor alleles, genotypes are indicated at the top of the panels. Denaturation, annealing, and elongation steps of PCR amplification are 30 s in duration each and were performed at 94, 60, and 72 °C, respectively, for a total of 35 cycles. The PCR product was visualized by electrophoresis on a 2% agarose gel.

**Table 2 jpm-12-01383-t002:** Summary of studies on the influence of the CYP2C19 genotype on the concentration of VCZ in pediatric patients.

Authors (Year of Publication)	Title	Objectives	Study Type	Results
Narita et al. (2013) [8]	Correlation of CYP2C19 Phenotype with VCZ Plasma Concentration in Children	Analysis of the metabolizer status as defined by CYP2C19 genotype and VCZ plasma concentrations	Retrospective study	VCZ Cmin higher in PMs and IMs than in NMs and UMs
Takahashi et al. (2021) [5]	CYP2C19 Phenotype and Body Weight-Guided VCZ Initial Dose in Infants and Children after Hematopoietic Cell Transplantation	Characterize the effects of CYP2C19 metabolizer status with covariateson the PK variability of prophylactic VCZ in pediatric patients after hematopoietic cell transplantation (HSCT)	Observational study	Dose to achieve target concentration: 33–50% lower for PMs, IMs 25–50% higher for RMs, UMs
Tian et al. (2021) [9]	Impact of CYP2C19 Phenotype and Drug-Drug Interactions on Voriconazole Concentration in Pediatric Patients	To study the key factors that affect VCZ Cmin in Chinese pediatric patients with hematological malignancies who have undergone HSCT	Retrospective study	IM: 0.31 mg/mL/mg/kg PM: 0.48 mg/mL/mg/kg (higher Cmin) EM: 0.11 mg/mL/mg/kg UM: 0.09 mg/mL/mg/kg Omeprazole: increased VCZ concentration Rifampicin: decrease VCZ concentration
Hicks et al. (2020) [10]	Prospective CYP2C19-Guided Voriconazole Prophylaxis in Patients with Neutropenic Acute Myeloid Leukemia Reduces the Incidence of Subtherapeutic Antifungal Plasma Concentrations	Describe the implementation of a prospective quality improvement study to determine if a higher prophylactic voriconazole dosage of 300 mg twice daily for CYP2C19 rapid metabolizers reduces the incidence of subtherapeutic trough concentrations without increasing voriconazole-induced toxicities	Prospective study	RM: received increased starting doses UM: avoid VCZ NM, IM: received standard starting doses
Garcia-García et al. (2021) [11]	Experience of a Strategy Including CYP2C19 Preemptive Genotyping Followed by Therapeutic Drug Monitoring of Voriconazole in Patients Undergoing Allogenic Hematopoietic Stem Cell Transplantation	Provide information to individualize VCZ treatment in immunocompromised pediatric patients and compare the results with those of Hicks et al.	Analyses of pediatric patients preemptively tested for the CYP2C19 genotype	Starting dose changes in 29% of patients NMs and IMs: received standard starting doses RM and UM: received increased initial doses No PM in the cohort
Wang et al. (2021) [12]	Model-Oriented Dose Optimization of Voriconazole in Critically Ill Children. Antimicrobial Agents Chemother	To use a PK model to optimize voriconazole dosing regimen in children with critical illness	Pharmacokinetic modeling study	30–40% lower maintenance doses in PM compared to EM
Zhao et al. (2021) [6]	Factors Affecting Voriconazole Trough Concentration and Optimal Maintenance Voriconazole Dose in Chinese Children	Investigate maintenance dose to optimize VCZ therapy and factors affecting trough VCZ concentration	Non-interventional retrospective clinical study	CYP2C19 genotype influenced VCZ Cmin
Chen et al. (2022) [13]	Combined Effect of CYP2C19 Genetic Polymorphisms and C-Reactive Protein on Voriconazole Exposure and Dosing in Immunocompromised Children	Identify factors associated with VCZ concentrations and doses required to achieve therapeutic concentrations	Retrospective study	NM: lower VCZ exposure and high daily dose needed to achieve the therapeutic concentration compared to PM Influence of other factors on VCZ concentration, such as C reactive protein

CYP2C19: cytochrome 2C19; EM: extensive metabolizer; IM: intermediate metabolizer; NM: normal metabolizer; PK: pharmacokinetic; PM: poor metabolizer; UM: ultra-rapid metabolizer; VCZ: voriconazole.

**Table 3 jpm-12-01383-t003:** Summary of studies on the influence of CYP2C19 genotype on PPIs.

Authors (Year of Publication)	Title	Objectives	Study Type	Results
Bernal et al. (2019) [20]	CYP2C19 phenotype and risk of proton pump inhibitor associated infections	To test the hypothesis that CYP2C19 metabolizing groups are associated with infectious events in children on PPIs	Retrospective cohort study	NM: higher infection rate compared to RM/UM
Franciosi et al. (2018) [19]	Association between CYP2C19*17 allele and pH probe testing in children with symptomatic gastroesophageal reflux	Investigate if PPI drug-resistant GERD may be related to CYP2C19 variants	Retrospective cohort study	CYP2C19*17 carriers: longer duration of exposure to an acidic pH
Mougey et al. (2019) [21]	CYP2C19 and STAT 6 variants influence the outcome of PPI Therapy in Pediatric Eosinophilic Esophagitis	Investigate the influence of the CYP2C19 genotypes on the treatment of eosinophilic esophagitis by PPIs	Prospective study	CYP2C19*17 carriers: insensitive to PPIs

CYP2C19: cytochrome 2C19; GERD: gastroesophageal reflux disease; NM: normal metabolizer; PPIs: proton pump inhibitors; RM: rapid metabolizer; STAT6: signal transducer and activator of transcription 6; UM: ultra-rapid metabolizer.

**Table 4 jpm-12-01383-t004:** Summary of studies on the influence of the *CYP2C19* genotype on antidepressants.

Authors (Year of Publication)	Title	Objectives	Study Type	Results
Ariefdjohan et al. (2021) [24]	The utility of pharmacogenetic guided psychotropic medication selection for pediatric patients: a retrospective study	Describe trends and clinical experiences in the application of pharmacogenetic testing in pediatric patients with neuropsychiatric disorders	Retrospective study	- 40% gene–drug interactions related to CYP2C19 metabolizer status - More side effects due to CYP2C19 metabolizer status with Paroxetine (SSRI) and Duloxetine (ISRN)
Strawn et al. (2019) [25]	CYP2C19 guided escitalopram and sertraline dosing in pediatric patients: a pharmacokinetic modeling study	Assess the impact of CYP2C19 metabolizer status on exposure to SSRIs (escitalopram or sertraline)	Pharmacokinetic modeling study	PM: higher Cmax and AUC
Aldrich et al. (2019) [26]	Influence of CYP2C19 metabolizer status on escitalopram/citalopram tolerability and response in youth with anxiety and depressive disorders	Investigate the association between CYP2C19 metabolizer status and response to antidepressant treatment	Retrospective study	PM: more adverse effects due to escitalopram/citalopram
Rossow et al. (2020) [27]	Pharmacogenetics to predict adverse events associated with pediatric antidepressants	To determine the association between the CYP2C19 genotype and the risk of side effects of (es)citalopram	Retrospective study	NM: higher adverse effects due to sertraline and escitalopram (surprising effect due to physiological differences in adolescence)

AUC: area under curve; Cmax: maximum concentration; CYP2C19: cytochrome 2C19; ISRN: selective noradrenaline reuptake inhibitors; MN: normal metabolizers; MP: poor metabolizers; SSRIs: selective serotonin reuptake inhibitors.

**Table 5 jpm-12-01383-t005:** SNPs and their corresponding oligonucleotide sequences. Destabilizing mismatches and allele-specific nucleotides (major/minor allele) are marked in bold and underlined.

Allele Name/SNP Position	SNP Identifier	Allele-Specific Primers—Forward	Allele-Specific Primers—Reverse
CYP2C19 *2 681	rs4244285	CACTATCATTGATTATT**A**CCC**G/A**	CTCCATTTTGATCAGGAAGC
CYP2C19*3 636	rs4986893	GGATTGTAAGCACCC**G**CTG**G/A**	AGAACTTTGCCATCTTTTCCAG
CYP2C19*17 806	rs12248560	GTGTCTTCTGTTCTC**T**AAG**C/T**	CAAATGGGAAAAGGGAGAC

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
