# Peer review of "Pharmacogenetic Expression of CYP2C19 in a Pediatric Population"

_jpm, 2022, doi:10.3390/jpm12091383_

Round 1
Reviewer 1 Report
I have reviewed the manuscript “Pharmacogenetic expression of CYP2C19 in pediatric population” a systematic review by Pierre-François et al submitted for publication in the Journal of Personalized medicine (MDPI).
My comments and questions to the authors are given below:
The manuscript is aimed to provide extensive knowledge of CYP2C19 gene variations and their impact on the therapeutic outcome among the pediatric population (age less than 21 years).
This review addresses one of the important contexts of the CYP2C19 pharmacogenetics among the pediatric population.
In this review, the authors aimed to elaborate on the impact of the CYP2C19 variations among voriconazole, antidepressants, proton pump inhibitors, and clopidogrel treated pediatric population with the evidence of 25 research articles from PubMed with specific criteria.
The overall review design and flow of the manuscript need a lot of improvement. I will list the things that must be improved in the review below.
1. In the review no introduction is provided about the CYP2C19 (even the gene acronym was not expanded). The authors should write about the CYP2C19 enzyme and gene location and what are the major drugs metabolized by it, and how many of them are prescribed for pediatric patients. Followingly the genotypes and the metabolizer pattern and how FDA provided guidelines to screen this variant before prescribing the drugs. This kind of start will provide a strong interest to the readers of the review.
2. The structure of the review should not mandatorily have an introduction, materials and methods, results, and discussion. I suggest authors rewrite the review as an introduction that emphasizes the CYP2C19 and followed by information about the literature reviewed and the logic of selecting the PubMed journals for the review. Followed by a discussion of individual drugs and explaining of evidence from reviewed manuscripts on how CYP2C19 genotype can impact the treatment efficacy in pediatric patients.
3. The Results section needs a lot of editing, I suggest the authors remove the heading Results and keep it as discussion and remove bullet points and numberings in each paragraph and it is not in proper order.
4. Another big concern in this review: In any systematic review, multiple papers addressing the issue will be compiled to explain the context in a single paragraph. In the current review, the authors explain each paper from PubMed as a single paragraph explaining the context of the individual paper. Which does not sound like a systematic review. For example, the flow of the discussion about the impact of CYP2C19 in Voriconazole should be as follows.
About the drug (Voriconazole) and its mode of action, what are the conditions used for therapy. How it is metabolized and how its metabolism is impacted by CYP2C19 genotypes. What is the current treatment dosage for pediatric patients referring to multiple papers (authors can point out if there is a difference between countries or if they are similar)? Then the effect of the genotypes in the trough concentration among pediatric patients with the evidence of multiple studies, how they are similar or how they differ. With all the discussion the authors should provide a conclusive remark as this might be the effective dosage for this CYP2C19, if it is inconclusive with the current literature evidence, the authors can emphasize it, so many future research can be focused to address this issue.
5. Like what I mentioned in the above point, the authors should rewrite the entire part for Clopidogrel, PPI, and antidepressants. A minimal introduction about the drugs, mode of action, metabolism by CYP2C19, and how often it’s prescribed for pediatrics will help the readers to understand the impact of the review.
6. “See Table-# summarizing the studies cited above” This sentence does not sound like professional writing; I suggest the authors write “refer table-#) in the middle of the discussion in every drug that will be more professional.
7. When authors write a review based on 25 PubMed research articles, do not restrict them to citing references from other manuscripts/reviews to explain about the pharmacology of the drugs and other information relevant to the review. I suggest the authors' review and include more manuscripts to bring the systematic review as an excellent knowledge source for the readers.
Reviewer 2 Report
Given that genetic variability in CYP2C19 may be associated with both lack of efficacy and toxicity of drugsdepending on its phenotypic expression, the authors of this review aimed at summarizing current knowledge relative to the association or treatment adaptation based on CYP2C19 genetics in pediatric population receiving drugs metabolized by this isoform such as voriconazole, antidepressants, proton pump inhibitors, clopidogrel. They also presented one of the approaches that they developed for detection of variant alleles in CYP2C19 gene. They found 25 articles on PubMed including pediatric patients (age < or equal to 21 years) having benefited from an assessment of CYP2C19. The results show that CYP2C19 poor and intermediate metabolizers exhibit a higher plasma concentration of voriconazole and PPIs compared to the rapid and ultra-rapid metabolizers.
There are several aspects:
- Defining a pediatric population that with an age <21 years is arbitrary since in many countries legal age is at 18. It would be good to perform sensitivity analyses with diverse cut-offs of age.
- There is no description of the search methodology. If this is a systematic review specific requirements should be met.
- The results are redundant with too much information reported in the text. The authors should make a better use of the Tables.
- The genotypic assay is reported on the Discussion as an aside but there too few methodological data (which should be in the methods). Either the authors expand this section or remove it from the paper.
Reviewer 3 Report
The manuscript entitled "Pharmacogenetic expression of CYP2C19 in pediatric population" is a well-conducted review, finely structured, and easy to read.
I have some comments:
1. I believe the Materials and Methods section could be improved. Please present the exclusion criteria and the search terms used in the research.
2. I think it would be of great value to include more information from PharmGKB on CYP2C19 (number of variants registered, level of confidence, drugs already studied). There is even the possibility of including only pediatric data.
3. Minor point: citing the gold standard diagnosis in the Genotyping assay section would be polite.
4. Minor point (2): I understood this is a narrative review based on systematic reviews and meta-analyses. However, the manuscript I received is categorized as a Systematic Review.
